# Gut Microbiota Analysis and In Silico Biomarker Detection of Children with Autism Spectrum Disorder across Cohorts

**DOI:** 10.3390/microorganisms11020291

**Published:** 2023-01-22

**Authors:** Wenjuan Wang, Pengcheng Fu

**Affiliations:** 1School of Life and Pharmaceutical Sciences, Hainan University, 58 Renmin Avenue, Haikou 570228, China; 2State Key Laboratory of Marine Resource Utilization in South China Sea, Hainan University, 58 Renmin Avenue, Haikou 570228, China

**Keywords:** human gut microbiota, autism spectrum disorder, ASD prediction, random forest, biomarker mining

## Abstract

The study of human gut microbiota has attracted increasing interest in the fields of life science and healthcare. However, the complicated and interconnected associations between gut microbiota and human diseases are still difficult to determine in a predictive fashion. Artificial intelligence such as machine learning (ML) and deep learning can assist in processing and interpreting biological datasets. In this study, we aggregated data from different studies based on the species composition and relative abundance of gut microbiota in children with autism spectrum disorder (ASD) and typically developed (TD) individuals and analyzed the commonalities and differences of ASD-associated microbiota across cohorts. We established a predictive model using an ML algorithm to explore the diagnostic value of the gut microbiome for the children with ASD and identify potential biomarkers for ASD diagnosis. The results indicated that the Shenzhen cohort achieved a higher area under the receiver operating characteristic curve (AUROC) value of 0.984 with 97% accuracy, while the Moscow cohort achieved an AUROC value of 0.81 with 67% accuracy. For the combination of the two cohorts, the average prediction results had an AUROC of 0.86 and 80% accuracy. The results of our cross-cohort analysis suggested that a variety of influencing factors, such as population characteristics, geographical region, and dietary habits, should be taken into consideration in microbial transplantation or dietary therapy. Collectively, our prediction strategy based on gut microbiota can serve as an enhanced strategy for the clinical diagnosis of ASD and assist in providing a more complete method to assess the risk of the disorder.

## 1. Introduction

Autism spectrum disorder (ASD) comprises a large group of heterogeneous neuro-developmental disorders characterized by symptoms such as clumsiness, repetitive behavior, abnormalities in social interaction, and difficulties in speech and communication [1]. In 2011, the number of individuals with ASD reached 67 million worldwide. In the United States, 1 in 68 children had ASD in 2014, while the ratio increased to 1 in 45 in 2016 [2]. Similarly, there were more than 10 million individuals with ASD in China in 2016, of which more than 2 million were children [3]. These statistics indicate that ASD has turned into a global disorder with an increasing incidence. At present, almost all individuals with ASD need special care and education services, causing a significant burden on their families and society. In 2014, it was estimated that the cost of supporting an individual with ASD and intellectual disability was 2.4 million USD during their lifespan in the United States and 2.2 million USD in the United Kingdom, while the cost was 1.4 million USD for individuals with ASD without intellectual disability in both countries [4]. The estimated monthly cost of caring for individuals with ASD in Chinese families in 2014 was over RMB 5000, which did not cover parental productivity loss [5]. Although ASD has gradually attracted worldwide attention and become a worldwide public health issue, its pathogenesis and mechanisms are still unknown; hence, there is no complete cure for ASD. However, it is generally recognized that the earlier ASD symptoms are detected and the more scientific interventions and treatments are carried out, the better the prognosis. Therefore, the timely detection and accurate diagnosis of ASD symptoms are of great practical significance to help the individuals and their families to live normal lives.

At present, ASD symptoms cannot be diagnosed by a single test, such as oral swabs, blood tests, or urine tests. Current ASD diagnosis is based on symptoms, behaviors, medical history, and social function, which are heavily dependent on subjective judgment. With developments in the field of biology, many scientists have begun to seek biomarkers to support ASD diagnosis and detection. Biomarker exploration has mostly focused on the brain [6,7,8], nervous system [9,10,11,12,13], genetic traits [14,15,16,17,18,19,20,21], and the presence of certain metabolites [22,23,24,25,26,27], as it is generally believed that ASD results from genetic [14,15,16,17,18,19,20,28], environmental [29,30], neurological, and immunological factors [31,32]. Some recent studies have revealed that there is a close interaction between the host and its gut microbiota, and an imbalance of the gut microbiota can cause ASD symptoms for children [33]. Therefore, the differential characteristics of gut microbiota between neurotypical individuals and those with ASD have emerged as a new type of potential biomarker for ASD diagnosis. Although the specific causal relationship and communication mechanism between the gut microbiota and ASD are still uncertain, many studies support the postulation for the existence of the microbiome–gut–brain axis [34,35,36,37]. It is believed that certain factors in the gut microbiota, such as key metabolites and cytokines, may affect the development of the central nervous system in some way, leading to mental disorders such as ASD.

It is seen that when the homeostasis of gut microbiota in individuals with ASD is disrupted, the differential diversity and abundance in their microbial compositions are observed, as shown in Table 1. Several researchers have attempted to diagnose ASD using certain phenotypes of gut microbiota as possible biomarkers, and to further treat ASD by manipulation of gut microbiota. A summary of these studies is shown in Table 1. Although these studies have confirmed that the gut microbiota of individuals with ASD exhibited abnormalities, there were no consistent conclusions on which microbial species in the gut microbiota were responsible for ASD. The difficulties were found to relate to several factors, such as small sample sizes, a single cohort design, and the disunity of research models in data mining. Therefore, more refined and effective strategies are needed to assess the risk, detection, and prediction of ASD via biomarker-based methods.

Traditional statistical analyses are seen to be unable to clarify correlations with human diseases due to the extreme complexity of the gut microbiota in its composition and function, including its variety, quantity, and complicated interactions. Alternatively, an increasing number of studies have adopted artificial intelligence, such as ML techniques, to assist in the diagnosis and prediction of human diseases, such as preterm birth [48], glycemic responses [49], Vibrio cholera infection [50], alcoholic hepatitis [51], colorectal carcinoma [52], and even biological age [53] and death [54]. These prediction models have achieved good performance, showing the substantial potential of artificial intelligence techniques in the analysis of gut microbiota for human health.

In terms of ASD diagnosis, several studies have been conducted to develop prediction models with satisfactory accuracy. Among them, Bosl et al. [8] used ML algorithms to predict the possibility of an infant with autism based on standard electroencephalography (EEG), which records the electrical activity of the brain. In a group of 188 infants aged about 9 months, the model achieved an accuracy rate of over 80% in differentiation of infants at high risk of autism from normal infants. Hazlett et al. [7] developed a deep learning algorithm to predict the status of children with high autism risk at the age of 24 months, utilizing the surface area information of brain magnetic resonance images of individuals at 6 and 12 months of age. This method achieved an accuracy of 81% and a sensitivity of 88%. Compared with the traditional behavior questionnaire with an accuracy of 50%, the reliability of the results was greatly improved. However, the prediction was limited to individuals with high familial risk and might not be able to guarantee the same effect in general cases. In addition, both models were applied to brain monitoring, which might be of limited use in follow-up treatments due to the complexity of the brain and our limited understanding regarding their functions. More recently, there appears a new trend to investigate the human diseases with the aid of cross-cohort analyses. Some of such studies across cohorts have shown that individuals from different racial and ethnic groups possess remarkable differences in gut microbiota [47,55,56]. It enables researchers to find similarities or differences in gut microbiota present in individuals with ASD from different regions and of different races, and to manipulate the gut microbiota toward that of healthy people by changing the diet of patients or transplanting the gut microbiota, to achieve the purpose of treating ASD.

Due to the heterogeneity of the population, high costs, and difficulty of data acquisition, most current studies on ASD are confined to single cohort data and the number of samples is usually not sufficient, which makes the clinical implementation of microbial-based diagnostic tools challenging. This work aims to explore the characteristics of gut microbiota of individuals with ASD across regions and look for potential biomarkers of ASD, to assist in ASD diagnosis. It integrates and annotates multi-source ASD data from different countries, cohorts, and ethnicities to provide a unified processing procedure. The commonalities and differences in the composition of gut microbiota of individuals with ASD across cohorts are explored to provide a more comprehensive and robust assessment of the correlation between ASD and gut microbiota. In addition, ML technology is used to establish an ASD prediction model and to determine potential biomarkers in silico to provide a novel method to assess the risk of ASD. Our results illustrate the AI’s role in the interpretation of gut microbiome for the prediction of ASD and suggest that a focus on biomarkers in the gut microbiota could be helpful in diagnosing ASD in the future.

## 2. Materials and Methods

### 2.1. Gut Microbiome Data Acquisition

To obtain comprehensive information on gut microbiota, we employed shotgun metagenomic sequencing data from NCBI in our analysis. To avoid possible biased results by inconsistent data processing procedures, we selected the primitive sequencing metadata-sequence read archive (SRA) rather than the microbial classification data from existing research platforms. The SRA approach is able to compare the datasets from original sequencing data to enhance the reproducibility. To minimize errors by simple sequencing technology, two independent SRA datasets with similar sequencing and procedures (PRJNA516054 [28] and PRJEB23052 [24]) were selected. A total of 51 typically developed individuals (TD group) and 73 individuals with a clinical diagnosis of ASD (ASD group) from Moscow city in Russia and Shenzhen city in China were aggregated (Table 2).

### 2.2. Microbiome Bioinformatics

As SRA data are a kind of non-text data, and cannot be analyzed directly, readable FASTQ files were extracted from the SRA file first. Secondly, the quality of the sequencing data in FASTQ files was evaluated. According to the quality control reports by FastQC [57], we used Trimmomatic [58] to obtain high-quality reads by cutting the adapters of the sequences TACACTCTTTCCCTACACGACGCTCTTCCGATCT and GTGACTGGAGTTCAGACGTGTGCTCTTCCGATCT and filtering the unsatisfied reads with the parameter of “LEADING:3 TRAILING:3 HEADCROP:4 SLIDINGWINDOW:4:15 MINLEN:36” for both the Moscow and Shenzhen cohorts. For the Shenzhen cohort, the adapter sequence GGAAGAGCGTCGTGTAGGGAAAGAGTGTAGATCTCGGTGGTCGCCGTATCATTAAAAAA was also removed. Second, we used Bowtie2 [59] to remove human reads by aligning the sequence to the hg38 build of the genome index. MetaPhlAn2 [60] was utilized to obtain the microbiome information for each sample, including the kingdom, phylum, class, order, family, genus, and species, as well as the corresponding abundances. In addition, all separated results from each sample were merged using MetaPhlAn2 merge_metaphlan_tables.py. Considering the interference of noise data, two criteria were designed to remove noise species, as described in Section 2.2. The results before and after modification were visualized using a Venn diagram.

### 2.3. Microbiome Data Analysis

The similarity between microbial samples was calculated using PCoA based on the Bray–Curtis algorithm. The correlations among different species were visualized through a co-occurrence network using Gephi version 0.9.2 [61]. The Spearman correlation coefficient between pairwise species was calculated using the Corr.test from the Psych package. The FDR-corrected *p* values of less than or equal to 0.05 and R values greater than or equal to 0.2 were considered significant. LEfSe analysis [62] was performed to determine the differentially abundant species, using the online analysis module provided by the Galaxy platform (http://huttenhower.sph.harvard.edu/galaxy, accessed on 15 July 2021). Differentially abundant species between the ASD and TD groups were tested using the pairwise Wilcoxon rank-sum test with linear discriminant analysis. The threshold value of the logarithmic linear discriminant analysis score for discriminative features was set as 3.0 and *p* = 0.05.

### 2.4. Prediction

Traditional statistical assessments are limited in feature selection, biomarker discovery, and diagnosis detection; thus, a more intelligent method is needed. ML is a collection of data analysis techniques that aim to learn patterns from multidimensional datasets and build predictive models based on associations between the features of a given dataset. The process of ML itself is to find a set of optimal model parameters and convert the features in the input data into accurate predictions for labels. The RF algorithm in ML is commonly considered to be effective when the number of features greatly exceeds the number of samples. The main workflow of ML consists of three steps: processing input data, learning or training the underlying model, and making predictions using new data. RF, as one of the main branches of ML, naturally follows these three main steps:Processing input data. The input data of this paper are the microbial species and their relative abundance information from each sample generated after species annotation, where “features” are each microbial species and their relative abundance, and “labels” is the category of each sample, including neurotypical individuals and those with ASD.Learning or training model. This step is mainly to find the optimal parameters of the model by repeating the sub-steps of “parameter estimation,” “model performance evaluation,” and “error identification and correction”.Once the optimal parameters are determined in step 2, the model with the optimal parameters is used to predict with the new input data.

We used the stratified 10-fold cross-validation method to train and test the model, taking into account the imbalance in the sample number of groups. As shown in Figure 1, samples for the groups of TD_Moscow, ASD_Moscow, TD_Shenzhen, and ASD_Shenzhen were stratified in 10 folds so that each fold contained approximately the same proportion of samples as the original dataset (the ratio was 2:3:3:4 based on their size). All sample data in sequence were assigned numbers 1 to 10 and each sample received a single index. For each fold, the sample data with a same index were individually used as the test set while the rest of the sample data were used as the training sets. This design ensured that the samples were utilized in turn both as training data in 9 folds and test data in one fold. In the training process, the grid search method was used to choose the optimal parameter of the feature entry, which takes each feature as the entry for training in turn, while the parameter of tree depth was set to the default value of 500. In addition, all the error rates corresponding to each feature entry were compared, and the optimal feature entry, which corresponds to minimum error rates, was defined. For this, the tree depth was set to 100, 200, and 300; the maximum was set to 10,000; the minimum error rate was calculated and the corresponding tree depth was chosen as the optimal value. Thus, the prediction model was trained. In the test process, the “labels” of the true value of the test data indicated that the person was a neurotypical individual or that with ASD was removed first. Then, the “features” of the test data were input into the prediction model and the prediction values of “labels” for each sample in the test set were output. We compared all the prediction results (i.e., the prediction value of labels, neurotypical individuals, or those with ASD) with the correct label (i.e., the real value of labels, neurotypical individuals, or those with ASD) to calculate the overall accuracy of the entire different cross-validation fold and presented it as the AUROC value.

Stratified 10-fold cross-validation is a total of 10 calculations based on different combinations of data for the same sample data. This method not only made full use of the limited sample data, but also traversed the possibilities to prevent accidental results due to data combination. The AUROC metric summarizes the true-positive rate (TPR) and false-positive rate (FPR) for the unequal proportions of each outcome [63]. TPR indicates the probability of correctly judging ASD samples from all ASD samples and represents the sensitivity of the model; FPR indicates the probability of misjudging TD samples as ASD samples and represents the specificity of the model. In general, the larger the value of AUROC, the better the classification performance. A total of 100 repeats of stratified 10-fold cross-validation were run, in order to reduce the contingency of the model. In each repeat, grid search was used to choose the optimal parameters for the training process. An AUROC value was output for each repeat and, finally, five highest AUROC values and their corresponding models were singled out from the 100 repeats. For improving the prediction results, a mean decrease accuracy (MDA) method (named “importance”) embedded in the RF algorithm was used to calculate the importance of each “feature,” which indicated the contribution of each microbial species and their relative abundance to the prediction. The uninformative species and their relative abundance with the value of MDA less than or equal to 0 were removed, and the rest of the species and their relative abundance were utilized to repeat steps before as new sample data. Thus, five new models with the highest AUROC values and important species from every 100 runs were selected to run the next round. The iteration was repeated until little or no increase in the AUROC value was reached (AUROC value < 0.01), and the corresponding model was optimal. Using this iterative method not only reduced the accuracy loss caused by limited sample data but also reduced the contingency of the optimal model and provided almost unbiased performance. The interactive method is illustrated in Figure 2.

## 3. Results

### 3.1. Species Composition of Gut Microbiota

All acquired SRA data were with their adapter sequences cut, quality-trimmed, and filtered using Trimmomatic. After removing the human reads by alignment of the obtained reads to the hg38 build of the genome index using Bowtie2, MetaPhlAn2 was utilized to obtain the microbiome profiles (Appendix A). As expected, only 30–75% of the species for each sample were clearly identified because of the complexity and diversity of the gut microbiome, and the reference genome sequence constructed at present is far from complete. As shown in Figure 3A and Appendix A, a total of 749 kinds of gut microbes were annotated from the microbiome profiles of the Shenzhen and Moscow cohorts, among which 648 species were identified in the Shenzhen cohort and 602 species in the Moscow cohort. As a result, a slightly richer microbial diversity was shown in the Shenzhen cohort than in the Moscow cohort.

Based on the samples from Shenzhen, 555 microbial species were found in the ASD group (ASD_Shenzhen), and 514 species were found in the control group (TD_Shenzhen); 441 species were found in both the ASD and TD groups, accounting for 68.1%. In the samples from Moscow, 544 microbial species were found in the ASD group (ASD_Moscow), and 439 species were found in the control group (TD_Moscow), of which 381 species (63.3%) were found in both groups. This is in line with our current understanding of the composition of gut microbiota that absolute beneficial bacteria and absolute harmful bacteria account for a small proportion in the gut. The most abundant bacteria are opportunistic pathogens, which are affected by the environment and other bacteria and easily switch between being beneficial and harmful. Data from both cohorts showed that the ASD group had a greater diversity of microbes than the control group (Shenzhen: 555 vs. 514; Moscow: 544 vs. 439). However, in both the Shenzhen and Moscow cohorts, the ASD group and the control group shared more than 60% of the same microbes. It was seen that the ASD groups in the two cohorts contained a total of 574 species of microbes, of which 425 species were in common, accounting for 74.0%. Meanwhile, there were only 130 unique species in the Shenzhen cohort and 119 unique species in the Moscow cohort. There were 588 species of microbes in the control group of the two cohorts, among which 329 species were in common, accounting for 56.0%. Meanwhile, 177 species were unique to the Shenzhen cohort, and 82 species were unique to the Moscow cohort. This suggests that the microbes that cause ASD may be in common in terms of species, even though they are geographically different.

Data filtering was conducted to maintain the accuracy, by the method in literature [48]. Two criteria were designed to remove noises in this study: (1) species with abundance values greater than or equal to 0.01 in less than 5% of the samples, and (2) species with abundance values greater than or equal to 0.001 in less than 15% of the samples. The total species accepted were decreased from 749 to 285, with up to 256 species shared by the four subgroups (Figure 3B and Appendix A).

Phylum level analysis was performed after noise filtering. It is observed that the abundant phyla were Actinobacteria, Synergistetes, Firmicutes, Bacteroidetes, Proteobacteria, Fusobacteria, and Verrucomicrobia (Appendix A). Both the gut microbiota of the two cohorts showed that the two dominant phyla were Firmicutes and Bacteroidetes. Compared to the TD group, the ASD group showed a higher Firmicutes/Bacteroidetes ratio, which is in agreement with previous studies [24,42] (Figure 3C). At the genus level, the top ten abundant genera in both ASD and TD groups were Bacteroides, Faecalibacterium, Bifidobacterium, Eubacterium, Alistipes, Prevotella, Roseburia, Blautia, Ruminococcus, and Shigella (Figure 3D and Appendix A). In addition, the hierarchical heatmap (Figure 3E) indicated that the genus Intestinimonasc was more abundant in the TD group, while Prevotella, Coprococcus, and Sellimonas were more abundant in the ASD group. Notably, the results we obtained for Prevotella are inconsistent with those of a previous study [42].

At the species level, the top ten species with the relative abundance in the ASD group were Faecalibacterium prausnitzii, Bacteroides vulgatus, Bacteroides uniformis, Prevotella copri, Bifidobacterium pseudocatenulatum, Bacteroides fragilis, Bacteroides dorei, Bacteroides ovatus, Eubacterium rectale, and Alistipes putredinis, while the top ten abundant species in the TD group were Bacteroides vulgatus, Bacteroides uniformis, Faecalibacterium prausnitzii, Bacteroides dorei, Bacteroides fragilis, Bacteroides plebeius, Alistipes putredinis, Escherichia coli, Bifidobacterium longum, and Ruminococcus gnavus. As shown in Figure 3F, among the top ten species with the highest abundance, six species in the ASD and TD groups were the same. They were all species with high abundance, but their relative abundance was slightly different. This demonstrates once again that the high abundance of basic microbes remains common in both individuals with and without ASD.

### 3.2. The ASD Group Was More Heterogeneous than the TD Group

As there was little difference in species diversity between the ASD and TD groups, a principal coordinate analysis (PCoA) was performed to investigate the variation in the microbial communities in the datasets of the two cities based on the Bray–Curtis algorithm (Figure 4 and Appendix A). The PCoA analysis showed an obvious clustering between the two datasets (the sample in the Shenzhen cohort presented relatively left, while the sample in the Moscow cohort presented relatively right), which implied that geographic location was an important factor in the samples. In addition, the results from both the Shenzhen and the Moscow cohorts exhibited a similar tendency in that the microbiota composition of the ASD group was more heterogeneous than that of the TD group. The samples of neurotypical individuals were found to be more similar, while the samples of individuals with ASD were more diverse. The results are similar to those obtained by 16S rRNA gene sequence analysis in the literature [25]. However, it was found that the segmentation distance between the ASD and TD groups was not obvious for both cohorts, and there was no significant difference between the ASD and TD group samples. In addition, the values of PC1 and PC2 were only 0.1125 and 0.0893, respectively, indicating that the differences between the ASD and TD groups were not obvious. 

### 3.3. No Biomarker Was Observed in the Species with Low Abundance

The linear discriminant analysis effect size (LEfSe) for discovering high-dimensional biomarkers was utilized (*p* < 0.05, LAD score > 3). The LEfSe analysis revealed a significant increase in the abundance of the species Bacteroides cellulosilyticus, Eubacterium rectale, Eubacterium_sp_CAG_180, Bacteroides intestinalis, Roseburia faecis, Ruthenibacterium lactatiformans, Firmicutes_bacterium_AM55_24TS, Coprococcus eutactus, Megamonas funiformis, Lachnospira pectinoschiz, Firmicutes_bacterium_CAG_65, Megamonas hypermegale, and Eubacterium_sp_CAG_581 in the ASD group compared to that in the TD group. Meanwhile, the ASD group showed a significant decrease in the abundance of Veillonella parvula, Bacteroides_coprocola_CAG_162, Eubacterium siraeum, Eubacterium_sp_CAG_248, Eubacterium_sp_CAG_202, Bacteroide stercoris, Bacteroides plebeius, and Bacteroides dorei (Figure 5). LEfSe was performed again with the original 749 species before modification. It was found that the result was the same as that after de-noising. This suggested that the data modification reserved important information and this analysis generated no biomarkers for low-abundance species.

### 3.4. Correlations in the ASD Group Were More Complex than Those in the TD Group

Co-occurrence networks for the species from both ASD and TD groups were constructed based on significant Spearman correlations to explore potential relationships among the species within the gut microbial communities. The network (false discovery rate <0.05, rho ≥ 0.2) was then visualized using Gephi version 0.9.2. The network relationships of the ASD group (285 nodes, 6997 edges, Appendix A) were slightly more complex in comparison to those of the TD group (285 nodes, 6255 edges, Appendix A). It was observed that the proportion of positive correlations (blue curves; 4626, 66.11% of ASD group; 4290, 68.59% of TD group) and negative correlations (yellow curves; 2371, 33.89% of ASD group; 1965, 31.41% of TD group) remained almost the same, while the absolute values were slightly different. The positive correlations between the gut microbiota in both ASD and TD groups were far more frequent than negative correlations.

### 3.5. Prediction Model Based on Random Forest Algorithm

It was found from the characteristics of gut microbiota data that neither PCoA nor co-occurrence network analysis showed qualitative differences in the microbial community between the TD and ASD groups. In this study, a random forest (RF) [64] classifier model was built for the classification of gut microbiota that could be present in the ASD group at the species level, which features non-linear and multi-dimensional relationships. In this model, a nested integrative method was improved for stratified 10-fold cross-validation (Figure 1). The details are given in Materials and Methods. The Moscow cohort (30 ASD samples and 20 TD samples) and the Shenzhen cohort (43 ASD samples and 31 TD samples) were analyzed separately and together. Prediction performance was evaluated using the area under the receiver operating characteristic curve (AUROC) metric. The average AUROC value of the final five models was regarded as the prediction result of the RF model. For the Shenzhen cohort alone, the prediction result achieved a high AUROC value of 0.984 and accuracy of 97%, with only one round of the 100-iteration run. The model used an average of 39 species of the full set of 749 species as features (Appendix A). The result of the Moscow cohort had a poor average result of AUROC = 0.81 and accuracy = 67%, with a total of six rounds of the 100-iteration run. Eventually, the iterations became stabilized in the fourth round, with the optimal average feature set containing 41 species (Figure 6A–C and Appendix A). For the combination of the two cohorts, the optimal model was achieved in the third round. The average prediction result had an AUROC = 0.86, accuracy = 80%, and an average feature set of 67 species (Figure 6D–F and Appendix A). This supports our RF model based on the idea that relative abundance of the microbial community can be used to accurately predict the clinical diagnosis of ASD.

We also tested the independence of the models cross Moscow and Shenzhen cohorts. The prediction model was trained by the data from one cohort, and validated with the other. The results showed a poor independence of the models with low AUROC values (around 0.5). This is consistent with the previous study [65], which focused on prediction for colorectal cancer.

### 3.6. Potential Biomarkers of ASD Diagnosis

As we aimed to capture potential biomarkers (i.e., species) to support ASD diagnosis, an evaluation method of feature importance by the mean decrease accuracy embedded in the RF algorithm was employed and an optimal prediction was achieved. To make the feature set smaller, an iterative method was adopted to enable optimal predictions with iteration to select the important features of the current round. The results showed that five models for the Shenzhen cohort predicted 41, 46, 33, 41, and 33 species as important features, with the optimal average feature set of 39 species. In comparison, five models for the Moscow cohort predicted 48, 34, 46, 38, and 38 species as important features (Table 3), with an average number of 41 species. The combined cohorts outputted 59, 77, 56, 72, and 71 species as important features from the final five optimal models (Table 4), with the optimal average feature set of 67 species. Furthermore, from the five best models with higher AUROC values, 27, 46, and 67 species were determined in at least three models from the Shenzhen cohort, Moscow cohort, and their combination, respectively (Appendix A). These important species contribute most to prediction models and are considered potential biomarkers for autism disorders. However, when comparing the potential biomarkers of the Moscow and Shenzhen cohorts together and alone, only Eubacterium_sp_CAG_248 and Prevotella copri species were seen to appear across the three sets of data (Figure 7 and Appendix A). The result of partially overlapping potential biomarkers supported the interpretation of the poor independence of the prediction models mentioned in Section 3.5. This was probably due to regional differences, which has been mentioned in previous studies of the gut microbiome [66]. Therefore, the complexity of gut microbiota and those potential biomarkers may not be interpreted from the presence of single species but a combination.

## 4. Discussion

Homeostasis of metabolism maintained by gut microbiota is not only important for host nutrition and viability, but also for human health and avoidance of disease; disturbed gut microbiota is believed to be a cause of many mental disorders, including ASD. To explore the characteristics of the gut microbiota of individuals with ASD across regions, and look for potential biomarkers of ASD to assist in ASD diagnosis, we attempted to aggregate and annotate multi-source human ASD data from different countries, cohorts, and ethnicities with a uniform processing standard treatment and obtain harmonious and comparable data across studies. It is clear that the gut microbiota is highly complex, which requires high-level and in-depth analysis to explore its system mechanisms. Traditional microbial analysis is not able to distinguish the neurotypical individuals and those with ASD. It is, thus, justified to explore ML technology to enable accurate diagnosis from the gut microbiota in individuals with ASD.

We established a prediction model using a RF algorithm based on the relative abundance of 285 species from two datasets of gut microbiota. The prediction results showed that the RF model possessed prediction power even with only dozens of microbial species. The accuracy of our models is comparable to other prediction models [7,8] based on brain monitoring data, while our method could be more practical for exploring the follow-up treatments due to the simplicity of regulating gut microbiota. A significant variation across the two cohorts is also indicated (Figure 7), which is consistent with the previous work of obvious regional clustering by other analysis (showed in Figure 4). Some other studies shown in Table 1 achieved positive results based on 16S rRNA gene sequencing data. These results also supported that microbial information would serve as a promising diagnosis tool of ASD. However, due to different samples and processing methods, there exists room for the consistency of the potential biomarkers for autism disorders to be improved, which indicates that the data were specific and should be normalized with a standard. In our results, the likelihood of Eubacterium_sp_CAG_248 and *Prevotella copri* was highest. Here, *Prevotella copri* is one of the dominant species in the ordinary human gut microbiome [67], while some studies [25,68] have shown that the relative abundance of *Prevotella copri* in the ASD group was seen to significantly decrease in comparison to the TD group. On the other hand, our LEfSe analysis and RF prediction suggested that the relative abundance of Eubacterium_sp_CAG_248 was remarkably different for ASD and TD groups. This is in agreement with previous studies in the literature. For example, a recent study [25] indicated that Eubacterium_sp_CAG_248 was associated with ASD. Recent studies have shown corrections between *Eubacterium* spp. and ASD or other mental diseases. *Eubacterium* spp. sampled from the vagina was reported to be with higher abundance in individuals with ASD [25]. In addition to ASD diagnosis, Eubacterium_sp_CAG_248, together with Eubacterium_sp_CAG_28 and Eubacterium_sp_CAG_86, were negatively associated with five phenotypes, including colorectal cancer, liver cirrhosis, inflammatory bowel diseases, type 2 diabetes, and atherosclerotic cardiovascular disease [69]. Dan et al. also showed that Eubacterium_sp_CAG_38 displayed a positive correlation with hexanoic acid level, while El-Ansary et al. [70] demonstrated that levels of acetic, valeric, hexanoic, and stearidonic acids in the blood were significantly higher in autistic patients. A study [71] indicated that Eubacterium_sp_CAG_202 and Eubacterium_sp_CAG_156 were 2 of the 29 depleted species in the patients with major depressive disorder. Thus, these reports were in support of our conclusion that Eubacterium_sp_CAG_248 and *Prevotella copri* were potential ASD biomarkers, although further investigation is needed. There still exist challenges for precise diagnostic assistance for ASD with the aid of ML technology, as the relationship between gut microbiota and ASD is so complex and unclear for accurate analysis at present. More comprehensive studies are, thus, required to understand which genes or metabolites contribute to the mechanisms of gut metabolism that contribute to biomarkers of ASD in gut microbiota.

To improve prediction ability, we constructed two types of deep learning models. One was a DNN classifier based on structured data of gut microbiota species and relative abundance, and the other was VGG net, based on the graphic data of gut microbiota species and relative abundance for each sample. Unfortunately, neither model achieved satisfactory results; both produced AUROC values worse than that of the RF model. This may be due to the insufficiency of the samples to support in these two deep learning models. It is well known that when the number of samples is less than the number of features, the RF algorithm possesses more advantages in accuracy. This result is consistent with several previous conclusions on the prediction of other diseases by using deep learning models. For example, Yuan’s team developed the DeepGene model based on the deep neural network [72], aiming to identify the types of cancer by learning the genetic mutation data of individuals with cancer. Although there were 3122 samples of 12 types of cancers and the gene number was 22834, the predicted AUROC value was only 0.6. This also indicates that the DNN classifier and VGG net models may need to be supported by massive data; otherwise, models that perform well given limited samples need to be developed. In the future, more samples from different cities need to be added to the prediction model. Given the scarcity of available data, we plan to increase the number of “neurotypical” samples with the assumption that gut microbiota for the TD group, even from different cohorts, is able to present similar characteristics as possible biomarkers. We believe that improvement of prediction capabilities of ML-based models is critical for the development of new strategies for smart ASD diagnostic assistance.

In conclusion, ASD prediction strategies based on gut microbiota could be used to assist the diagnosis of ASD, and to assess ASD risks. The findings in our work are of use to the development of novel ASD diagnosis and treatment procedures. In addition, the results of our cross-cohort analysis suggested that various influencing factors, such as population characteristics, geographic regions, and dietary habits, should be taken into consideration. In addition, biomarker detection could be sensitive to data collection and processing, which could become dataset-specific. Therefore, a large amount of standardized data with more factors, collected and processed with the same criteria, should be analyzed in silico to explore the potential for clinical practice. Although the results presented in this study are far from being used directly in the actual diagnosis of ASD, they serve as a starting point to inspire subsequent development and research, as well as provide a paradigm shift for the study of other human diseases associated with gut microbiota.

## Figures and Tables

**Figure 1 microorganisms-11-00291-f001:**
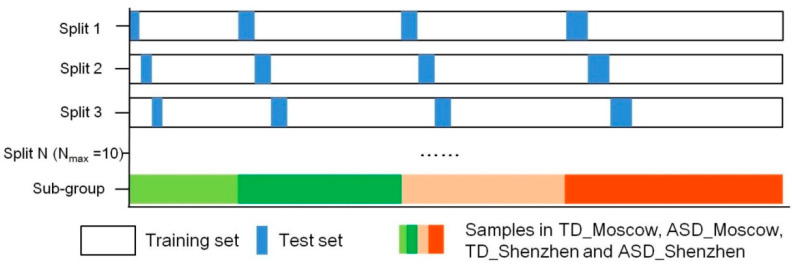
Stratified 10-fold cross-validation. All models were evaluated by stratified 10-fold cross-validation with the dataset divided into training and test sets.

**Figure 2 microorganisms-11-00291-f002:**
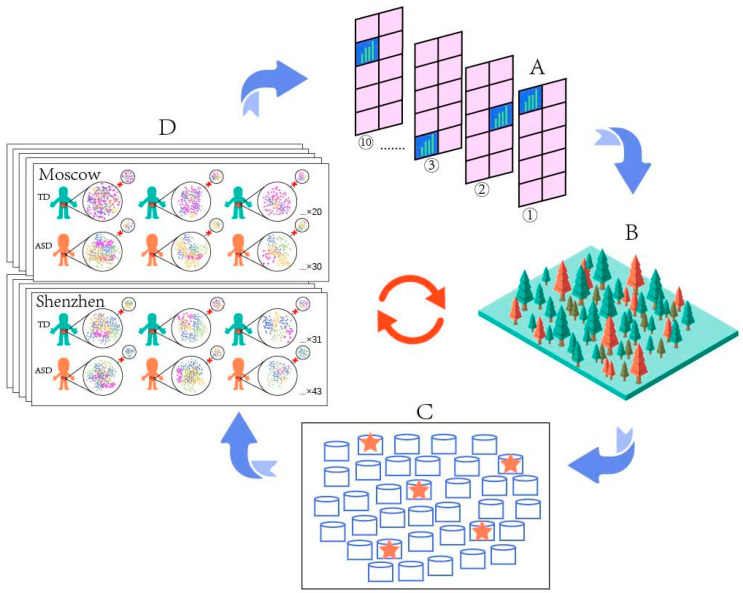
The main interactive processes of our random forest algorithm. (**A**) Stratified 10-fold cross-validation was conducted with a dataset divided into training and test sets. (**B**) A total of 100 repeats of stratified 10-fold cross-validation were run, and five models with the highest AUROC values were singled out (**C**). (**D**) After removing the unimportant species with a value of mean decrease accuracy less than or equal to 0, the rest of the species were utilized to run the next round of 100 repeats of stratified 10-fold cross-validation. The iteration was repeated until little or no increase in the AUROC value was reached (AUROC value < 0.01), and the corresponding model was optimal.

**Figure 3 microorganisms-11-00291-f003:**
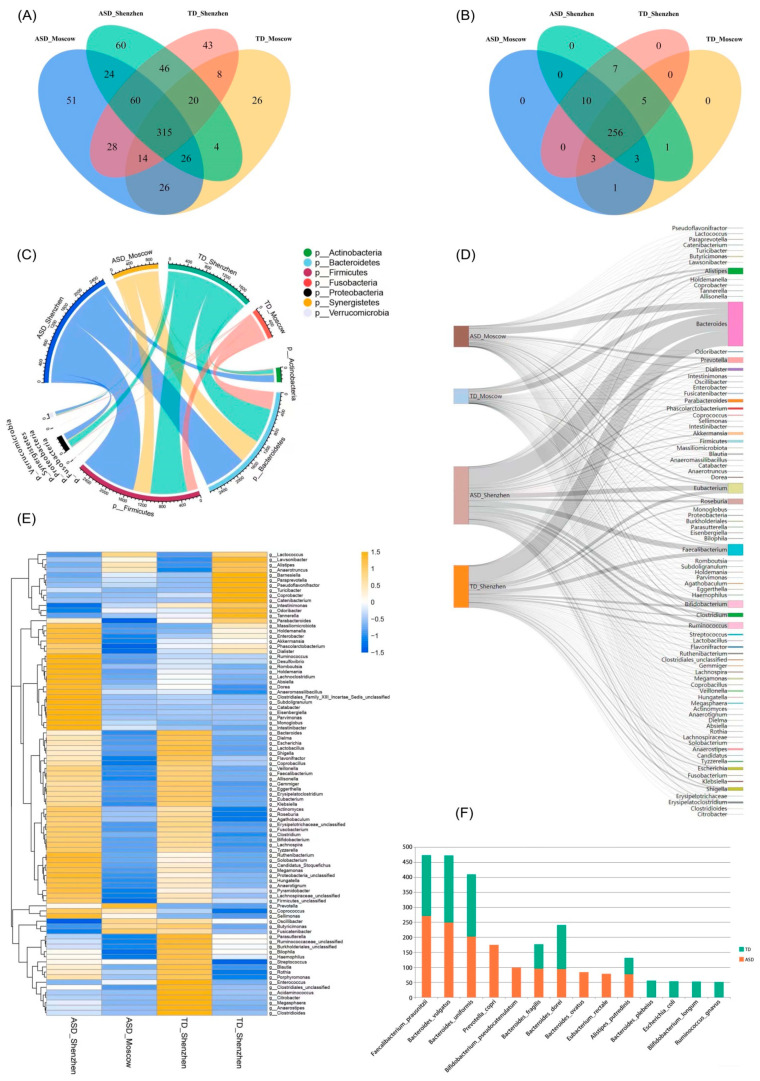
Microbiome profiles of ASD and TD groups. (**A**) There were a total of 749 species in the gut microbiota for our analysis. (**B**) After the data reduction to exclude those species with noise and less information, 285 species remained. (**C**) The most abundant phyla in ASD and TD groups. (**D**) The top ten abundant genera in both ASD and TD groups. (**E**) The hierarchical heatmap indicated the more abundant genus in the TD group and the more abundant genus in the ASD group. (**F**) Species composition of ASD and TD groups ranked among the top ten in relative abundance at the species level.

**Figure 4 microorganisms-11-00291-f004:**
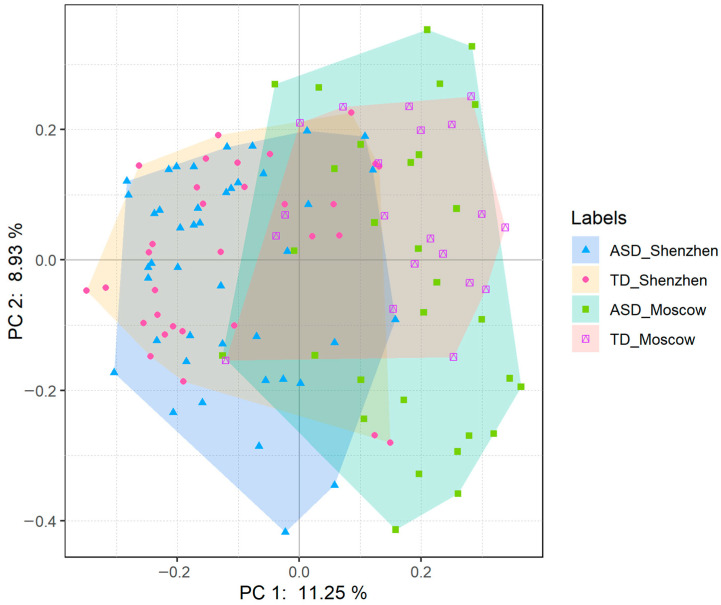
PCoA analysis based on Bray–Curtis algorithm. Results showed a clear clustering between two datasets and a similar tendency that the gut microbiota composition of the ASD group clusters was more heterogeneous than that of the TD group.

**Figure 5 microorganisms-11-00291-f005:**
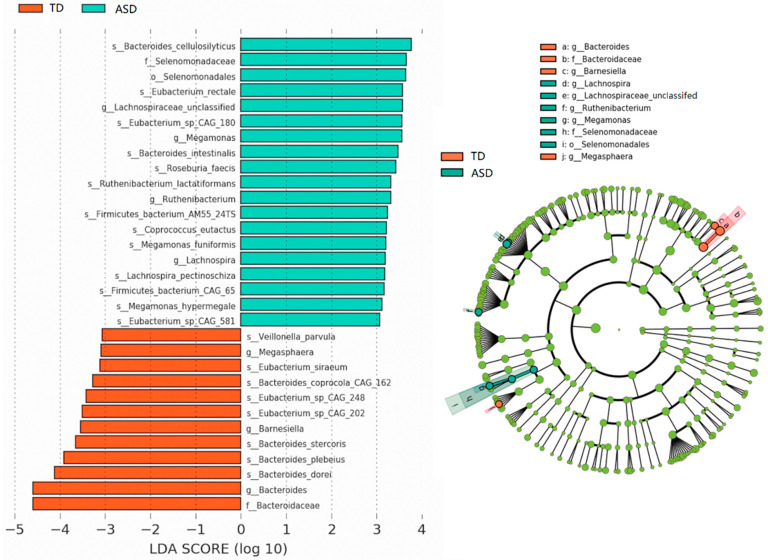
LEfSe analysis. The bar chart (**left**) shows the significantly different species in TD and ASD groups, and the circle chart (**right**) shows the taxonomic rank of the different species from phylum to genus.

**Figure 6 microorganisms-11-00291-f006:**
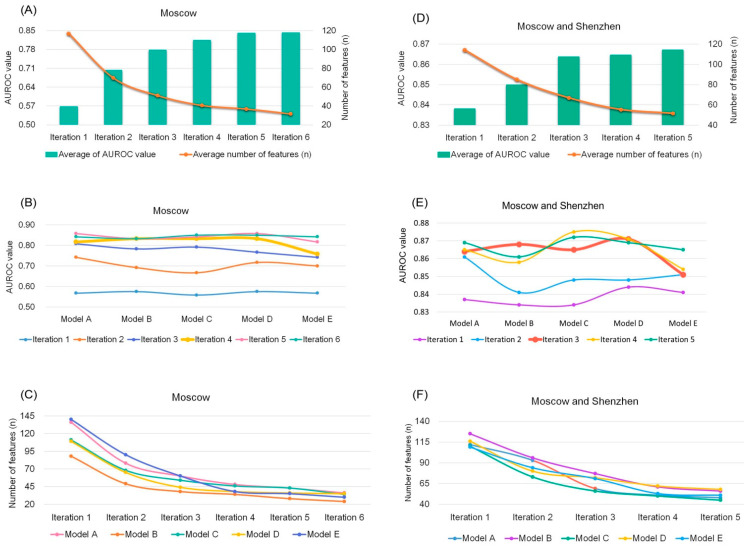
The results of the prediction model. The Moscow cohort (**A**–**C**) alone had a poor average result of AUROC = 0.81 and accuracy = 0.67, with a total of four rounds of 100 runs, and the optimal average feature set was 41 species. The Shenzhen cohort alone achieved a high value of AUROC = 0.984 and accuracy = 0.968 with just one round of 100 runs, and the optimal model used an average of 39 species of the full set of 285 species as features. For the combination of two cohorts (**D**–**F**), the optimal model was achieved in the third round, showing the average prediction result of AUROC = 0.86, accuracy = 0.80, and an average feature set of 67 species.

**Figure 7 microorganisms-11-00291-f007:**
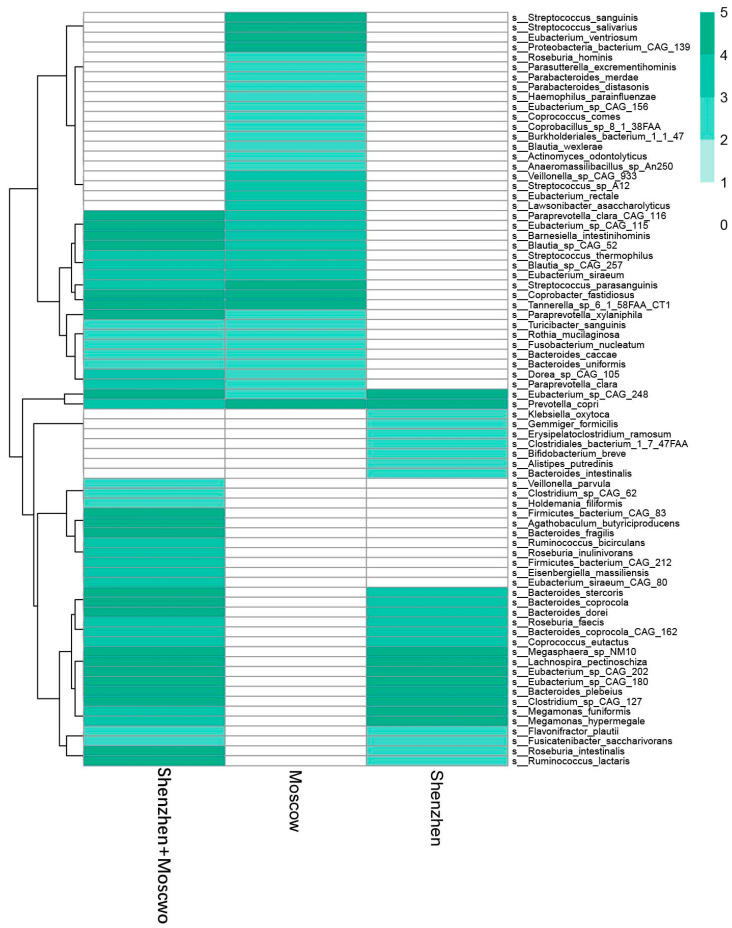
Potential biomarkers. Only Eubacterium_sp_CAG_248 and Prevotella copri species were present across the Moscow cohort, the Shenzhen cohort, and both cohorts combined; they were considered the potential biomarkers.

**Table 1 microorganisms-11-00291-t001:** Findings of previous studies. Differences in gut microbiota between individuals with autism spectrum disorder (ASD group) and typically developed individuals (TD group).

Model	Number of Samples	Country or Region	Sequencing Methods	Manifestation of Species Disorder (ASD)	Reference
children	TD: 10; ASD: 10	Slovakia	Realtime-PCR	phyla *Bacteroidetes*/*Firmicutes*: ↓ *; *Lactobacillus*: ↑; *Bifidobacterium*/*Lactobacillus*, *Streptococcus thermophillus*, the total bacteria content: -	[38]
children	TD: 45; ASD: 45	China	16S rRNA V3-V4	At phylum level: -; genera *Lachnoclostridium*, *Tyzzerella* subgroup 4, *Flavonifractor*, unidentified_Lachnospiraceae: ↓	[39]
children	TD: 20; ASD: 20	-	16S rRNA V2-V3	genera *Prevotella*, *Coprococcus*, and unclassified_Veillonellaceae: ↓	[40]
children	TD: 35; ASD: 6	China	16S rRNA V3-V4	phyla *Bacteroidetes*/*Firmicutes*; genera *Sutterella*, *Odoribacter* and *Butyricimonas*: ↑ genera *Veillonella* and *Streptococcuse*: ↓	[41]
children ASD	TD: 40; ASD: 40	Italy	16S rRNA V3-V5	phylum *Bacteroidetes*, genera *Alistipes*, *Bilophila*, *Dialister*, *Parabacteroides*, *Veillonella*: ↓; phyla *Firmicutes*/*Bacteroidetes*; genera *Collinsella*, *Corynebacterium*, *Dorea*, and *Lactobacillus*; *Escherichia*/*Shigella* and Clostridium cluster XVII; fungal: genus *Candida*: ↑	[42]
mice	TD: 10; ASD: 10	USA	16S rRNA V3-V5	classes *Bacteroidia*, *Clostridia*: ↑	[43]
children	TD: 3; ASD: 3	UK	FISH-FCM	phyla *Clostridium* spp.: ↑ *Bifidobacterial*: ↓	[44]
children	TD: 20; ASD: 18	USA	16S rRNA V4	genera *Bifidobacterium*, *Desulfovibrio*: ↓	[23]
mice	TD: 21; ASD: 25	Canada	qRT-PCR	phyla *Firmicutes*: ↓ *Bacteroidetes*: ↑	[45]
children and mothers	TD: 30; ASD: 59	China	16S rRNA V1-V2	Children: phylum Proteobacteria: ↑; genera *Enhydrobacter*, *Chryseobacterium*, *Streptococcus*, and *Acinetobacter*: ↑; species *Acinetobacter rhizosphaerae*, *Acinetobacter johnsonii*, *Prevotella melaninogenica*: ↓ Mother: families Moraxellaceae and Enterobacteriaceae, genus *Faecalibacterium*: ↓	[46]
minors	TD: 450 ASD: 569	China, Ecuador, Italy, Korean	16S rRNA V3-V4, V4, V4-V5	Results were variable according to different analysis methods and parameter settings.	[47]
children	TD: 31 ASD: 43	China	Shotgun metagenomic sequencing	phylum *Actinobacteria*: ↑; *three Clostridium* taxons, two *Eggerthella* taxons, two *Klebsiella* taxons: ↑; taxons *Bacteroides vulgatus*, *Betaproteobacteria*, *Campylobacter jejuni* subsp. *jejuni* 81–176, *Campylobacter jejuni* subsp. *jejuni* ICDCCJ07001, *Candidatus* Chloracidobacterium thermophilum B, *Coraliomargarita akajimensis* DSM 45221, *Proteus mirabilis*, and HI4320 *Spirochaeta thermophila* DSM 6192: ↓	[24]
children	TD: 20 ASD: 30	Moscow	Shotgun metagenomic sequencing	species *Enterococcus faecium*, Megasphaera *elsdenii, Bacteroides fragilis:* ↑	[28]

*: ↓: Significant decrease; ↑: Significant increase; -: no significant change.

**Table 2 microorganisms-11-00291-t002:** Description of sample data.

Characteristic	Moscow Cohort	Shenzhen Cohort
Subjects of ASD (n)	30	43
Subjects of TD (n)	20	31
Age (years)	3–5	2–7
Sequencing instruments	Illumina NovaSeq 6000	Illumina HiSeq 4000
Layout	PAIRED	PAIRED
AvgSpotLen	300	300
Bytes (Gb)	1.92–4.08	0.526–4.09

**Table 3 microorganisms-11-00291-t003:** The number of species as important features in the five optimal prediction models selected in each iteration of the Moscow cohort. The iterations became stabilized in the fourth round.

	Model A	Model B	Model C	Model D	Model E	Average
1st iteration	136	88	111	109	140	117
2nd iteration	78	49	68	65	90	70
3rd iteration	60	38	54	44	60	51
4th iteration	48	34	46	38	38	41
5th iteration	43	28	43	36	35	37
6th iteration	36	24	34	35	30	32

**Table 4 microorganisms-11-00291-t004:** The number of species as important features in the five optimal prediction models selected in each iteration of the Moscow and Shenzhen cohorts. The iterations became stabilized in the third round.

	Model A	Model B	Model C	Model D	Model E	Average
1st iteration	112	125	110	116	109	114
2nd iteration	93	96	73	80	84	85
3rd iteration	59	77	56	72	71	67
4th iteration	51	61	50	62	53	55
5th iteration	48	56	45	58	51	52

## Data Availability

The prediction code was written in R and is available online at https://github.com/dubi77/RF.git, accessed on 10 October 2022.

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
