# Peer review of "Gut Microbiota Analysis and In Silico Biomarker Detection of Children with Autism Spectrum Disorder across Cohorts"

_microorganisms, 2023, doi:10.3390/microorganisms11020291_

Round 1

Reviewer 1 Report

Wang et al. analyzed data from different studies based on the species composition and relative abundance of gut microbiota in children with ASD and TDs, established a predictive model to explore the diagnostic value of the gut microbiome for children with ASD, and identified the potential biomarkers for ASD diagnosis. This is an exciting study, and the manuscript is well-written. The author needs to address the concerns before being accepted for publication.

1. cross-cohort validation. The predicted model was built based on Moscow and Shenzhen cohorts. To build up confidence, the authors need to develop the model based on Moscow and Shenzhen cohorts separately and predict Shenzhen and Moscow to access the model-independent cross-cohort accuracy.

2. The author needs to compare their results to published studies to discuss the convergence and differences.

Author Response

We are very sorry to reply lately due to the Covid infection. We appreciate your constructive comments and suggestions. Please see the attachment.

Reviewer 2 Report

The authors presented an impressive study on a predictive model using  an machine learning algorithm to explore the diagnostic value of the gut microbiome for the children with ASD  and identify potential biomarkers for ASD diagnosis.

Although I don't have much to comment on the study design I suggest the authors to spend more in describing the possibile future application of their result in the clinical practice. 

Author Response

(The authors gave the same response as above.)
